**PLOS** NEGLECTED TROPICAL DISEASES

# Trichinella spiralis cathepsin L damages the tight junctions of intestinal epithelial cells and mediates larval invasion

**Ruo Dan Liu, Xiang Yu Meng, Chen Le Li, Xin Zhi Lin, Qiu Yi Xu, Han Xu, Shao Rong Long, Jing Cui\*, Zhong Quan Wang\***

Department of Parasitology, Medical College, Zhengzhou University, Zhengzhou, PR China

\* cuij@zzu.edu.cn (JC); wangzq@zzu.edu.cn (ZQW)

## Abstract

### Background

Cathepsin L, a lysosomal enzyme, participates in diverse physiological processes. Recombinant *Trichinella spiralis* cathepsin L domains (rTsCatL2) exhibited natural cysteine protease activity and hydrolyzed host immunoglobulin and extracellular matrix proteins in vitro, but its functions in larval invasion are unknown. The aim of this study was to explore its functions in *T. spiralis* invasion of the host's intestinal epithelial cells.

### Methodology/principal findings

RNAi significantly suppressed the expression of TsCatL mRNA and protein with TsCatL specific siRNA-302. *T. spiralis* larval invasion of Caco-2 cells was reduced by 39.87% and 38.36%, respectively, when anti-TsCatL2 serum and siRNA-302 were used. Mice challenged with siRNA-302-treated muscle larvae (ML) exhibited a substantial reduction in intestinal infective larvae, adult worm, and ML burden compared to the PBS group, with reductions of 44.37%, 47.57%, and 57.06%, respectively. The development and fecundity of the females from the mice infected with siRNA-302-treated ML was significantly inhibited. After incubation of rTsCatL2 with Caco-2 cells, immunofluorescence test showed that the rTsCatL2 gradually entered into the cells, altered the localization of cellular tight junction proteins (claudin 1, occludin and zo-1), adhesion junction protein (e-cadherin) and extracellular matrix protein (laminin), and intercellular junctions were lost. Western blot showed a 58.65% reduction in claudin 1 expression in Caco-2 cells treated with rTsCatL2. Co-IP showed that rTsCatL2 interacted with laminin and collagen I but not with claudin 1, e-cadherin, occludin and fibronectin in Caco-2 cells. Moreover, rTsCatL2 disrupted the intestinal epithelial barrier by inducing cellular autophagy.

### Conclusions

rTsCatL2 disrupts the intestinal epithelial barrier and facilitates *T. spiralis* larval invasion.

**Data Availability Statement:** All relevant data are included in the paper.

**Funding:** RDL was supported by the National Natural Science Foundation of China (81802025)

and Henan Province Science and Technology key project (212102310146); JC was supported by the National Natural Science Foundation of China (No. 82172300). The funders had no role in the study design, data collection and analysis, decision to publish, or preparation of the manuscript.

**Competing interests:** The authors have declared that no competing interests exist.

## Author summary

Trichinosis is a severe zoonotic disease that affects humans by eating raw or semi-raw meat products containing *T. spiralis* larvae. The intestinal mucosal barrier is the initial defense against *T. spiralis* infection and the main point of contact between the host and the parasite. Beyond the mechanical action, the protease released by the worm are essential for *T. spiralis* invasion of the host small intestine. Cathepsins play a crucial role in the migration of *Fasciola hepatica* and the invasion of *Schistosoma japonicum*, essential for their parasitism in the host. Nevertheless, the specific functions of *T. spiralis* cathepsin L during the worm's invasion of the intestinal mucosa remain unexplored. In this study, the TsCatL gene was silenced using siRNA, which inhibited worm invasion of Caco-2 cells, impaired worm development, and decreased female fertility. Incubation of rTsCatL2 with Caco-2 cells resulted in the translocation of cell tight junction proteins (claudin 1, occludin and zo-1), adhesion junction protein (e-cadherin) and extracellular matrix protein (laminin) into the cytoplasm or the nucleus, a reduction in claudin 1 expression and cellular autophagy, leading to disruption of the intestinal epithelial barrier. This study provides a novel target for developing anti-trichinosis vaccines and drugs.

## Introduction

Trichinosis is a severe foodborne zoonotic disease that affects humans who consume raw or semi-raw meat products containing *Trichinella spiralis* larvae [1]. Larval invasion into the host's intestinal mucosa for further development is essential in *T. spiralis* infection [2]. The intestinal mucosal barrier constitutes the primary natural defense against *T. spiralis* infection and serves as the main site of interaction between the host and the parasite. The intestinal mucosal barrier consists of mechanical, chemical, immune, and biological barriers. The mechanical barrier mainly comprises intestinal epithelial cells (IEC) and cell junctions. IEC junctions contain tight junctions (TJ), adhesion junctions (AJ), gap junctions (GJ), and desmosomes [3]. Intestinal epithelial TJ proteins comprise two major groups: transmembrane proteins and cytoplasmic proteins, primarily responsible for "barrier" and "defence" functions. Transmembrane proteins can regulate the transport of substances in the paracellular pathway, and cytoplasmic proteins mainly regulate the free diffusion of lipids and proteins. The TJ proteins mainly include claudin, occludin, zonula occludens protein 1 (zo-1) and β-catenin. The TJ proteins between cells close the apical spaces between epithelial cells and maintain the integrity of the intestinal mucosal barrier. The TJ proteins of epithelial cells are a dynamic barrier subject to regulation. Various endogenous or exogenous stimuli can compromise barrier integrity through diverse pathways, such as altering the content and distribution of TJ proteins, inducing cytoskeletal protein rearrangements, or modifying the phosphorylation of TJ proteins [4].

During *T.* spiralis invasion of the host small intestine, besides mechanical effect, the protease secreted by the worm also play a crucial role [5,6]. When *T. spiralis* larvae are co-cultured with IEC monolayers, the worms secrete a variety of proteases that damage the intestinal epithelium's integrity and make it easier for the worms to invade the IEC [7,8]. Serine protease of *T. spiralis* muscle larvae (ML) can reduce TJ proteins expression and disrupt the Caco-2 cell monolayers through the mitogen-activated protein kinase (MAPK) signaling pathway [9]. The expressions of occludin, claudin 1, and e-cadherin were downregulated, while claudin-2 was upregulated in the *T. spiralis* intestinal infective larvae (IIL) by cysteine and serine proteases

[10]. The expressions of occludin and claudin-3 in mouse intestinal epithelium were downregulated and claudin-2 expression was upregulated 2 days after *Trichinella* infection, which was associated with increased permeability of the small intestinal epithelium [11]. In addition, serine proteases produced by *Trichuris muris* can degrade the intestinal mucus barrier [12]. Giardipain-1, a cysteine peptidase produced by *Giardia duodenalis* trophozoites, causes the monolayer of intestinal epithelial cells to degrade the TJ proteins occludin and claudin 1 [13]. The *T. spiralis* cathepsin L (TsCatL) protein consists of a transmembrane helix, an inhibitor_I29 domain and a mature Pept_C1 domain. TsCatL two domains (TsCatL2) was cloned and expressed in *Escherichia coli* Rosetta-gami B (DE3) [14]. Recombinant TsCatL2 has cysteine protease activity and can hydrolyze host hemoglobin, serum albumin, immunoglobulin and extracellular matrix proteins in vitro with host-specific hydrolytic activity, indicating that TsCatL is a crucial digestive enzyme in *T. spiralis* [14]. In contrast, there are no relevant studies on the effect of TsCatL on epithelial barrier damage.

This study explored the role of TsCatL in invading the host's intestinal mucosa through RNA interference (RNAi) silencing of TsCatL in worms. Additionally, it investigated TsCatL's impact on disrupting the epithelial barrier through the interaction of rTsCatL2 with Caco-2 cells in vitro.

## Materials and methods

### Parasites, animals and ethics statement

*Trichinella spiralis* (ISS534) was sourced from pigs in Nanyang, Henan Province, and maintained in BALB/c. Specific pathogen free (SPF) female BALB/c, aged 4 to 6 weeks, were obtained from Huaxing Experimental Animal Center (No. SCXK 2019–0002). All animal experiments were approved by the Life Science Ethics Committee of Zhengzhou University.

### Preparation of ML and soluble protein

BALB/c mice infected by gavage with 300 *T. spiralis* were decapitated and killed after 42 days of infection. Skin, guts and fat were removed from the mice and added to a pepsin-hydrochloric acid artificial digestion solution and digested on a constant temperature shaker at 42°C for 4 ~ 6 h until no intact muscle was present, and ML were collected by natural sedimentation and a modified Baermann's method [15,16]. Soluble proteins were collected as follows: *T. spiralis* was first put through a grinder at 4°C for 1 min, followed by a 10 min sonication and centrifugation at 12,000 *g* for 30 min [7]. The supernatant was *T. spiralis* soluble protein. Protein concentration was determined using the BCA assay kit (Beyotime, China).

### Preparation of rTsCatL2, anti-rTsACatL2 serum and experimental cells

The pMAL-c2X/TsCatL2 recombinant plasmid were stored in our laboratory. Recombinant proteins (rTsCatL2) and MBP were expressed in *Escherichia coli* Rosetta-gami B (DE3), purified with amylose resin, and detected in previous studies [14]. To eliminate endotoxin interference in rTsCatL2, endotoxin removal resin (Thermo Fisher, USA) was used. Mouse anti-rTsACatL2 serum, anti-rTsASP2 serum and anti-MBP serum were produced by immunizing BALB/c mice with rTsCatL/rTsASP2/MBP mixed with adjuvant via intramuscular injection and collecting the blood of the mice to isolate the serum [6,14]. The human colonic epithelial cell line Caco-2 and the mouse myogenic cell line C2C12 were obtained from the Cell Resource Centre, Chinese Academy of Sciences.

**Table 1. TsCatL-specific siRNA and control-siRNA sequences.**

| siRNA name | Sense (5′-3′) | Antisense (5′-3′) |
|---|---|---|
| TsCatL siRNA-302 | GAAAUAUACGGAAAAACGUUTT | ACGUUUUUCCGUAUAUUUCUT |
| TsCatL siRNA-775 | CGCUUUUGAGUAUGUCAAATT | UUUGACAUACUCAAAAGCGUT |
| TsCatL siRNA-1077 | CAUUGAAAGGAAAGGAUUAUT | UAAUCCUUUCCUUUCAAUGTT |
| Control siRNA | AUCGGCUACCAAGUCAUACTT | GUAUGACUUGGUAGCCGAUTT |

### Electroporation of *T. spiralis* ML with siRNA

Three small interfering RNA (siRNA) sequences targeting TsCatL were designed using the siDirect version 2.0 online tool (http://sidirect2.rnai.jp). A control siRNA unrelated to *T. spiralis* was used as a control (Table 1). All siRNAs were synthesized by Sangon Biotech (Shanghai, China). The RNA interference (RNAi) experiment was divided into 5 groups: siRNA-302 group, siRNA-775 group, siRNA-1077 group, control-siRNA group and PBS group.

 *T. spiralis* ML were washed with sterile PBS (more than 16 times) and then resuspended in electroporation buffer containing 5 μM siRNA. Electroporation was performed at 125 V for 25 ms using a Gene Pulse Xcell System (Bio-Rad, USA). Subsequently, the larvae were incubated in 1640 medium at 37°C in a 5% $CO_2$ for 2 h [6]. Fetal bovine serum was added to the medium cultured for 1 ~ 5 days. Each experiment was done in triplicate.

### Transcription and expression of TsCatL2 after siRNA transfection

*T. spiralis* ML RNA was extracted using trizol (Tiangen, China), reverse transcribed into cDNA (PrimeScriPt RT reagent kit, Takara, Japan), and the qPCR reaction system was configured using SYBR Green qPCR Master Mix (Topscience, China). The qPCR amplification of TsCatL and glyceraldehyde-3-phosphate dehydrogenase (GAPDH) was performed using a 7500 Fast Real-time PCR System (Applied Biosystems, USA). The qPCR primers for TsCatL2 gene were 5′-TACGGAAAAACGTATGCAAATG-3′ and 5′-CAAATTCTCCATGAGTCAAA TCGG-3′. Specific primers for the GAPDH gene (GenBank: AF452239) were 5′-AGATGCTC CTATGTTGGTTATGGG-3′ and 5′-GTCTTTTGGGTTGCCGTTGTAG-3′ [17]. Finally, the relative quantitative $2^{-\Delta\Delta Ct}$ method was employed to analyze mRNA transcript levels of TsCatL at different time points following siRNA interference [18].

 20 μg soluble proteins from the worms were subjected to 10% SDS-PAGE and western blot. Depending on the molecular weight of the proteins, the membranes were incubated with mouse anti-rTsCatL2 serum (1:100), mouse anti-rTsASP2 serum (1:100) and rabbit anti-GAPDH IgG (1:1000) at 4°C overnight, then incubated with HRP-conjugated secondary antibodies (1:5 000, proteintech, USA) and exposure with the efficient chemiluminescence (ECL) solution (meilunbio, China) [19]. And then, the relative intensities of each band were analyzed using the Image J software (National Institutes of Health, USA).

### *T. spiralis* invasion assay in vitro

Caco-2 and C2C12 cells were grown into dense monolayers in 12-well or 12-well plates with slides. *T. spiralis* ML were incubated in 5% bile saline for 2 h at 37°C. 500 μL of 50°C 1.75% sterile low melting point agarose was mixed with 500 μL of 37°C 1640 medium containing about 50 live or dead worms and rapidly added to the cell surface. Once the semi-solid medium had solidified, the 12-well plates were incubated for 2 h, and the larval invasion into the cells was observed [20]. The semi-solid medium was then removed and the cells were stained with 10 μg/mL propidium iodide (PI) solution on ice for 1 min. After PBS washing, the damage to the cells caused by the worm invasion was observed under the microscope.

Subsequently, the cell surface was fixed with 4% paraformaldehyde fixative and incubated with anti-rTsCatL2 serum, *Trichinella*-infected mouse serum and normal mouse serum, respectively. FITC-conjugated goat anti-mouse IgG was added and incubated for 1 hour at 37˚C. The residues of rTsCatL on the cell surface were observed under a fluorescence microscope [21].

In addition, the anti-rTsCatL2 serum, anti-MBP serum, *Trichinella*-infected serum, normal mouse serum or PBS was added to the semi-solid medium at different dilution ratios to analyze the effect on larval invasion and inhibition. The effect of TsCatL silencing on the invasion rate of worms was examined by adding TsCatL siRNA-302, control-siRNA and PBS electro-transformed worms to a semi-solid medium containing a monolayer of Caco-2 cells, respectively.

## Animal experiments analyze the effect of TsCatL on larval development and survival

Ninety BALB/c mice were randomly assigned to the TsCatL siRNA-302, control siRNA and PBS group. The electrotransformed larvae were cultured in 1640 medium for 3 d, and then 200 *T. spiralis* ML were gavaged into each mouse. IIL and adult worms (AW) were obtained from the mice's intestines at 24 h and 6 d post-infection and counted and measured [17,22]. The 6d females were placed in 48-well plates for a further 48 h and the newborn larvae (NBL) generated by each female were counted and measured. At 35 d after *Trichinella* infection of the mice, the ML was obtained from mouse muscle and the number of worms per gram of muscle was calculated.

## Effect of rTsCatL2 on Caco-2 cell junctions

Caco-2 and C2C12 cells were grown into dense monolayers on glass coverslips in 12-well plates. rTsCatL2, MBP, E64-treated rTsCatL2, heat-treated rTsCatL2 or PBS was acidified in pH 4.5 buffer for 30 min, added to pH 6.5 PBS to adjust the final concentration to 10 μg/mL, then added to the surface of Caco-2 cells and incubated in a 5% $CO_2$ incubator for various times (1 ~ 60 min) [10,13]. Cellular immunofluorescence assays were performed by sequentially fixing the incubated slides with 4% paraformaldehyde for 10 min, antigen repair buffer (100 mM Tris, 5% [w/v] urea, pH 9.5) for 10 min at 95˚C, permeabilizing PBS (0.25% Triton X-100) for 10 min, and then incubating the slides with blocking solution (1% BSA, 22.52 mg/mL glycine PBST) for 30 min. After blocking, primary antibodies, including mouse anti-rTsCatL2 serum, mouse anti-MBP serum, rabbit anti-claudin 1, e-cadherin, occluding, zo-1, fibronectin or laminin antibodies (Abcam, USA) were added and incubated for 60 min. A 1:100 dilution of FITC-goat anti-mouse IgG (Proteintech, USA) or FITC-goat anti-rabbit IgG (Proteintech, USA) was incubated for 60 min [23]. Finally, the PI staining solution was added to stain nuclei for 5 min and the sections were embedded in 30% glycerol for observation under a fluorescence microscope.

Changes in cellular protein expression were detected by collecting cells after incubation, extracting cellular proteins with radio immunoprecipitation assay (RIPA) lysate (Beyotime, China), determining protein concentrations and then performing SDS-PAGE and western blot. Depending on the molecular weight of the proteins, polyvinylidene fluoride (PVDF) membranes were incubated with rabbit anti-claudin 1, e-cadherin, occludin, laminin or collagen-I antibodies (Abcam, USA), and mouse anti-ß-actin antibodies (Servicebio, China) were incubated overnight at 4˚C, followed by incubation with HRP-goat anti-rabbit/mouse IgG (Proteintech, USA). Exposure was performed using the ECL chemiluminescence kit (Meilunbio, China).

## Co-IP

Caco-2 cells were incubated with 10 μg/mL rTsCatL2, MBP or PBS for 30 min. The cells were washed with PBS and lysed with inhibitor-containing RIPA lysate (Beyotime, China) to obtain cell-soluble proteins. After the removal of non-specific binding, 2 μg anti-rTsCatL2 IgG was added to the rTsCatL2 group, 2 μg anti-MBP IgG was added to the MBP group and normal mouse IgG was added to the PBS group and shaken slowly at 4°C overnight. The next day, 20 μL of Protein A/G PLUS agarose (1:100; Santa Cruz, USA) was added to each group and shaken at 4°C for 1 hour. The precipitate was washed 5 times with RIPA lysate, 20 μL 1 × protein loading buffer was added to the precipitate and boiled at 100°C for 5 min, followed by SDS-PAGE and western blot. After transferring the proteins to PVDF membranes, the protein bands were incubated with rabbit anti-claudin 1, e-cadherin, occludin, collagen I, fibronectin or laminin antibody to analyze the cellular proteins interacting with rTsCatL2.

## Effect of rTsCatL2 on autophagy

A final concentration of 10 μg/mL rTsCatL2, 10 μg/mL MBP, earle's balanced salt solution (EBSS) or pH 6.5 PBS was incubated with Caco-2 cells for 60 min. The treated cells were analyzed for the induction of autophagy in Caco-2 cells using the monodansylcadaverine (MDC) Method Cell Autophagy Assay Kit (Beyotime, China). The procedure was as follows: Each well received 500 μL of MDC staining solution at 37°C for 30 min. The MDC staining solution was removed and washed with assay buffer to visualize green fluorescence under a fluorescence microscope with ultraviolet (UV) excitation light.

Normal untreated cells and cells after rTsCatL2 incubation were fixed with 2.5% glutaraldehyde for 5 min. The cells were gently scraped off with a spatula and centrifuged at 2500 *g* for 2 min. The cells were fixed for 2 h at room temperature in a new electron microscope fixative. After fixation, the samples were sent to Wuhan Servicebio for embedding, sectioning and observation of autophagy using transmission electron microscopy and photography.

## Statistical analysis

In this study, SPSS 21.0 software was utilizied to describe and analyze experimental data using a one-way ANOVA or chi-square test, with a significance level set at $P < 0.05$.

# Results

## Effect of silencing the TsCatL gene

The qPCR results revealed that mRNA transcript levels of TsCatL were reduced in siRNA-302 and siRNA-775 treated groups at 1 day post-RNAi (dpr) ($F_{1d} = 26.25$, $P < 0.05$). At 3 and 5 dpr, TsCatL mRNA transcript levels were reduced in siRNA-302, siRNA-775 and siRNA-1077 treated groups ($F_{3d} = 14.31$, $P < 0.05$; $F_{5d} = 85.46$, $P < 0.05$), and transcript levels were reduced by 44.42%, 41.06% and 39.43% in the three treatment groups at 5 dpr, respectively (Fig 1A). Western blot results showed that at 3 dpr, TsCatL protein expression was reduced by 44.07% ($P < 0.05$) in the siRNA-302-treated group. At 5 dpr, TsCatL protein expression was reduced by 45.58% and 39.17% ($P < 0.05$) in the siRNA-302 and siRNA-775 treated groups, respectively (Fig 1B). The RNAi specificity assay showed a 41.93% ($P < 0.05$) reduction in TsCatL protein expression in the siRNA-302-treated group at 3 day post-RNAi. In contrast, TsASP2 protein expression in the worms remained unchanged, indicating that the siRNA-302 silencing effect was specific to TsCatL only (Fig 2).

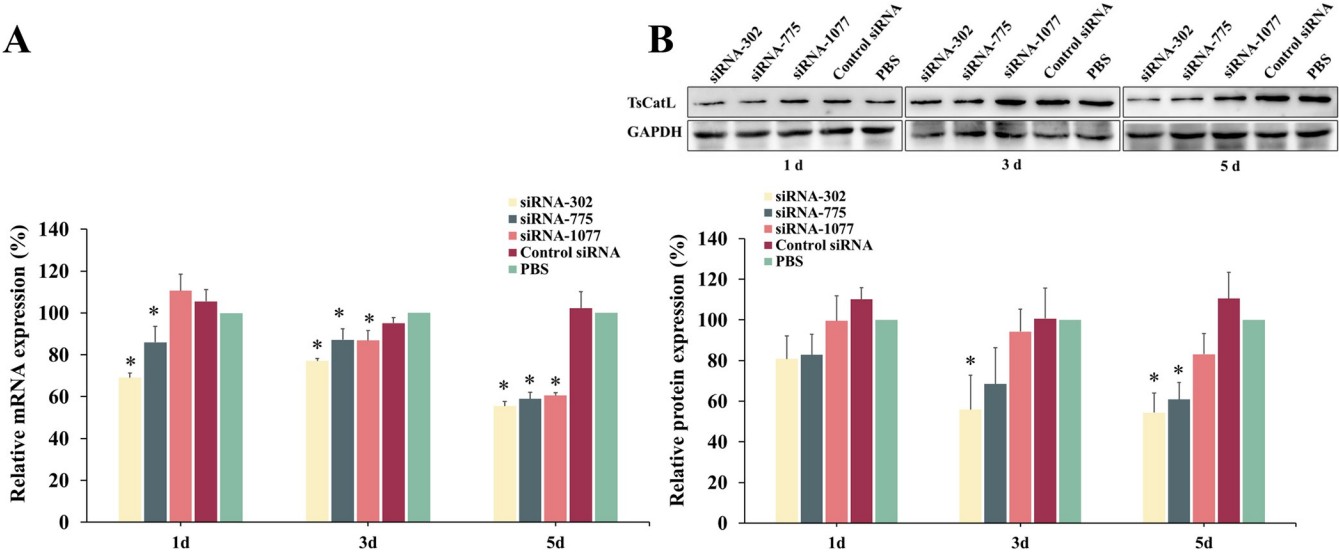

**Fig 1. Changes in the mRNA transcript levels (A) and protein expression levels (B) of the TsCatL gene after RNA interference.** A: qPCR analysis of TsCatL mRNA transcript levels; at 3 and 5 days post-RNAi, TsCatL mRNA transcript levels decreased in siRNA-302, siRNA-775 and siRNA-1077 treated groups (*$P < 0.05$). B: Western blot analysis of TsCatL protein expression. At 5 days post RNAi, TsCatL protein levels reduced by 45.58% and 39.17% in siRNA-302 and siRNA-775 treated groups, respectively (*$P < 0.05$).

## Effect of TsCatL in larval invasion

*T. spiralis* ML were added to the surface of different cell monolayers in vitro after bile activation, and some of the live worms were able to invade Caco-2 cell monolayers, leaving serpentine migration tracks on the surface of Caco-2 cells without invading C2C12 cells (Fig 3A). PI staining of the cells showed that the damaged cells were distributed in a serpentine pattern overlapping with the worm migration tracks (Fig 3B). Addition of rTsCatL2 immune serum to the surface of the cells showed a distinct bright green fluorescence of damaged cells, indicating that TsCatL2 remained in Caco-2 cells after larval invasion (Fig 3C). When anti-rTsCatL2 serum (1:50 ~ 1:200) was added to the medium, the invasion rate of worms into Caco-2 cell monolayers was 31.67%, 33.67% and 33.33%, respectively, which was lower than the invasion rate of 52.67% in the PBS group ($\chi^2_{1:50} = 9.023$, $P < 0.05$; $\chi^2_{1:100} = 7.334$, $P < 0.05$; $\chi^2_{1:200} = 8.160$, $P < 0.05$); the inhibition rate of 1:50, 1:100 and 1:200 anti-rTsCatL2 serum was 39.87%, 36.07% and 36.72%, respectively, which was lower than the normal serum ($\chi^2_{1:50} = 65.864$, $P < 0.001$; $\chi^2_{1:100} = 63.943$, $P < 0.001$; $\chi^2_{1:200} = 67.249$, *$P < 0.001$) (Fig 4A). Furthermore, in the TsCatL siRNA-302, control siRNA and PBS groups, the invasion rate of *T. spiralis* was 32.67%, 50.33% and 53.00%, respectively. The inhibition rates for the siRNA-302 group and the control siRNA group were 38.36% and 5.04%, respectively. The siRNA-302 group inhibited larval penetration into the Caco-2 cell monolayer more than the control siRNA group ($\chi^2 = 32.262$, $P < 0.001$) (Fig 4B). According to the aforementioned findings, TsCatL facilitated *T. spiralis* to invade the host epithelial cells.

## RNAi silencing of TsCatL on worm infectivity and fecundity

ML were orally infected in mice 3 d after siRNA treatment, and 24 h IIL, 6 d AW and 35 d ML were collected and then counted, observed and measured for length under a microscope. The recoveries of IIL and AW in the TsCatL siRNA-302 group were lower than those in the control siRNA and PBS groups ($F_{IIL} = 177.34$, $F_{AW} = 246.67$, $P < 0.05$), and the reduction rates of IIL

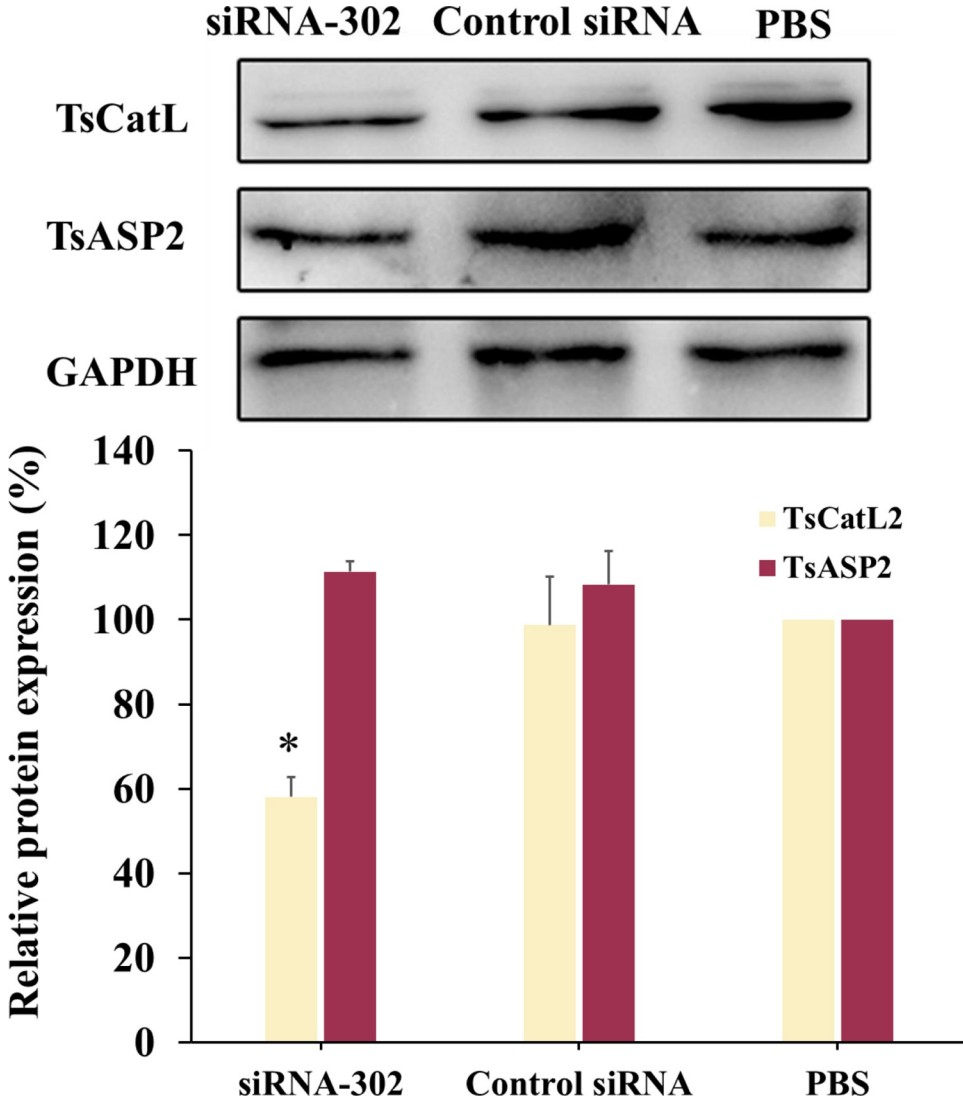

**Fig 2. Gene-specific analysis of RNA interference.** Western blot showing changes in TsCatL and TsASP2 protein expression levels in siRNA-302-treated worms (*$P < 0.05$).

and AW were 44.37% and 47.57%, respectively. Silencing of TsCatL reduced *T. spiralis* invasion into the intestine of mice and reduced the worm burden in the intestine of mice. The recovered 6 d females were incubated in 1640 complete medium for 24 h, and the number of NBL produced by females in the TsCatL siRNA-302 group was lower than that in the control siRNA and PBS groups ($F_{NBL} = 6.342$, $P < 0.05$). When the ML was collected in mice, the TsCatL siRNA-302 group had a lower worm burden than the control siRNA and PBS groups ($F_{ML} = 14.421$, $P < 0.05$) and a worm reduction rate of 57.06%. Length measurements of worms collected at different developmental stages revealed differences in the length of 6 d females between treatment groups ($F = 4.326$, $P < 0.05$) and 6 d females in the TsCatL siRNA-302 group were shorter than PBS group ($P < 0.017$), while the lengths of IIL, 6 d males, NBL and ML did not differ between treatment groups ($F_{IIL} = 1.306$, $P > 0.05$; $F_{Male} = 2.118$, $P > 0.05$; $F_{NBL} = 1.419$, $P > 0.05$; $F_{ML} = 2.772$, $P > 0.05$) (Fig 5).

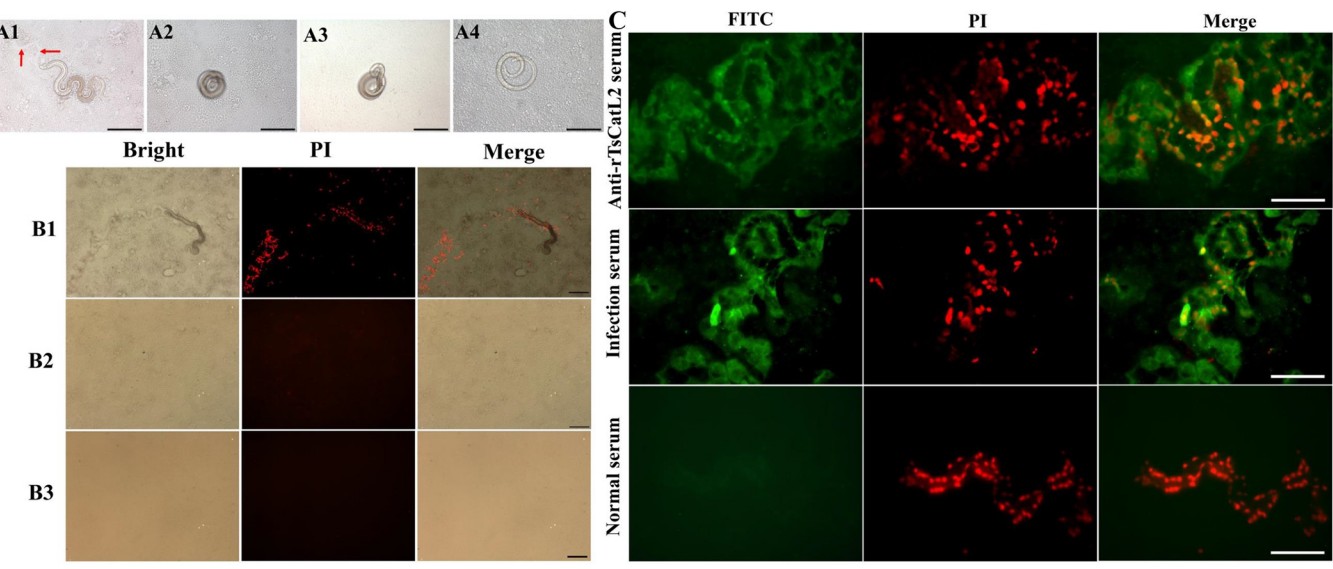

**Fig 3. Caco-2 cells damage caused by larval penetration.** A: Observation of larval invasion cell. The larva invaded Caco-2 cells (A1, the red arrow shows migratory trace); non-invaded larvae on Caco-2 (A2) and C2C12 (A3); the dead larvae on Caco-2 (A4); Scale bar: 200 μm. B: Larval invasion resulting in cell damage. B1 shows Caco-2 cells monolayer damaged by helminth invasion, stained with PI to show the red fluorescence of the cells; B2 represents undamaged Caco-2 cell monolayer; B3 shows undamaged C2C12 cell monolayer; scale bar: 200 μm. C: Fluorescence analysis of larval migrationin Caco-2 cells. After the larvae have invaded the cell monolayer, some worm proteins remain within the cells. Anti-rTsCatL2 serum and *T. spiralis* infection serum fluoresce green in association with worm proteins; nuclei of damaged or dead cells are stained red with PI; scale bar: 100 μm.

## Fluorescence detection of rTsCatL2 binding to Caco-2 cells

When rTsCatL2 was added to Caco-2 cells, the cells gradually shrank and became rounded with increasing incubation time. The tight junctions between the cells disappeared and some of the cells even fell off. In contrast, the cell morphology of the MBP and PBS groups did not change. Immunofluorescence with anti-rTsCatL2 antibody showed that rTsCatL2 bound mainly around the cell periphery at the beginning and gradually entered the cytoplasm and nucleus with increasing incubation time; when MBP and E64-treated rTsCatL2 was added to Caco-2 cells, faint fluorescence of Caco-2 cells was detected; whereas no fluorescence was seen in the heat-treated rTsCatL2 group and the PBS group (Fig 6).

## Fluorescence detection of the effect of rTsCatL2 on Caco-2 cell proteins

When rTsCatL2 was added to the surface of Caco-2 cells, immunofluorescence showed that claudin 1, e-cadherin, occludin and zo-1 were initially located at the tight junctions. After 30 min, the cells became smaller and separated from each other, claudin 1, occludin and zo-1 became smaller in outline and wrapped around each cell, and the cells became less connected to each other. However, e-cadherin aggregated into clusters and was engulfed inside the cells. After 60 min, immunofluorescence showed that claudin 1, e-cadherin, occludin and zo-1 entered the nucleus of the cells, the fluorescence of claudin 1 had disappeared in some cells, the fluorescence of e-cadherin, occludin and zo-1 was still present, and occludin and zo-1 also started to cluster (Fig 7A). Furthermore, after 60 min incubation, immunofluorescence showed that fibronectin was located in the cytoplasm, while laminin was scattered in the nucleus (Fig 7B). While the localization of cellular proteins did not change in the MBP, E64-treated rTsCatL2, and heat-treated rTsCatL2 and PBS groups.

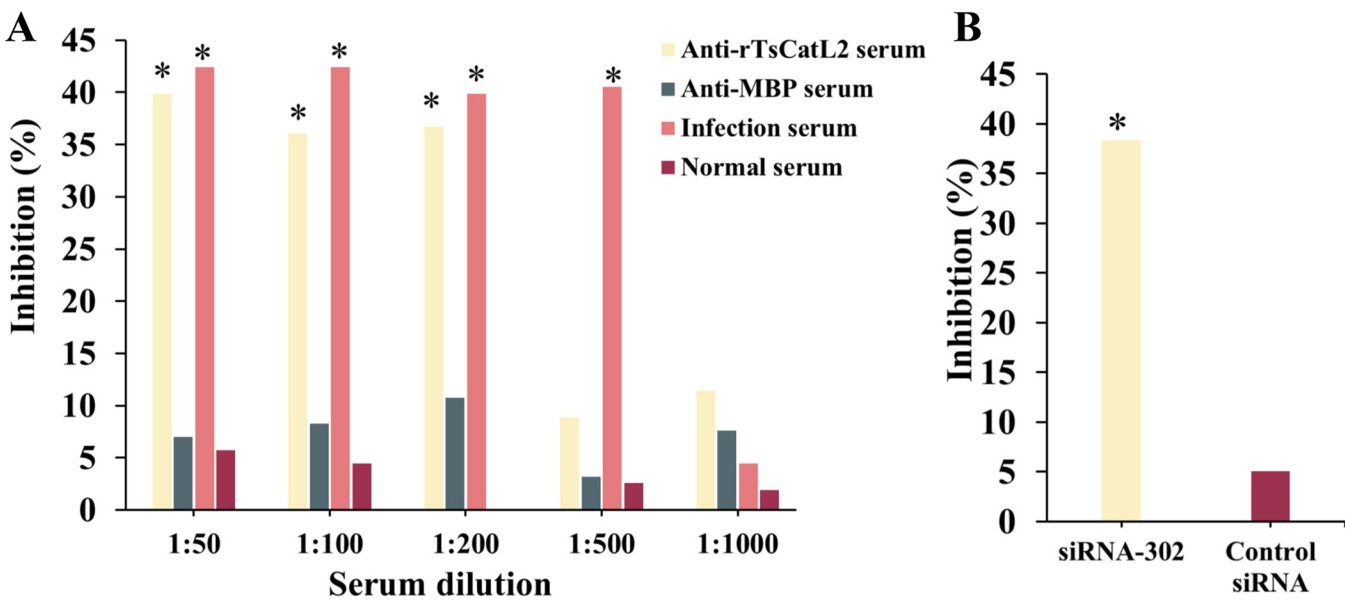

**Fig 4. The effect of TsCatL in the larval invasion of Caco-2 cells.** A: Larval invasion inhibition by anti-rTsCatL2 immune serum. After adding anti-rTsCatL2 serum at 1:50, 1:100 and 1:200 ratios, the inhibition rate was 39.87%, 36.07% and 36.72%, respectively, which was lower than the normal serum ($\chi^2_{1:50} = 65.864$, $P < 0.001$; $\chi^2_{1:100} = 63.943$, $P < 0.001$; $\chi^2_{1:200} = 67.249$, *$P < 0.001$). B: RNAi silencing of TsCatL inhibited larval invasion into Caco-2 cells. The inhibition rate of the siRNA-302 group was higher than that of the control siRNA group ($\chi^2 = 32.262$, $P < 0.001$).

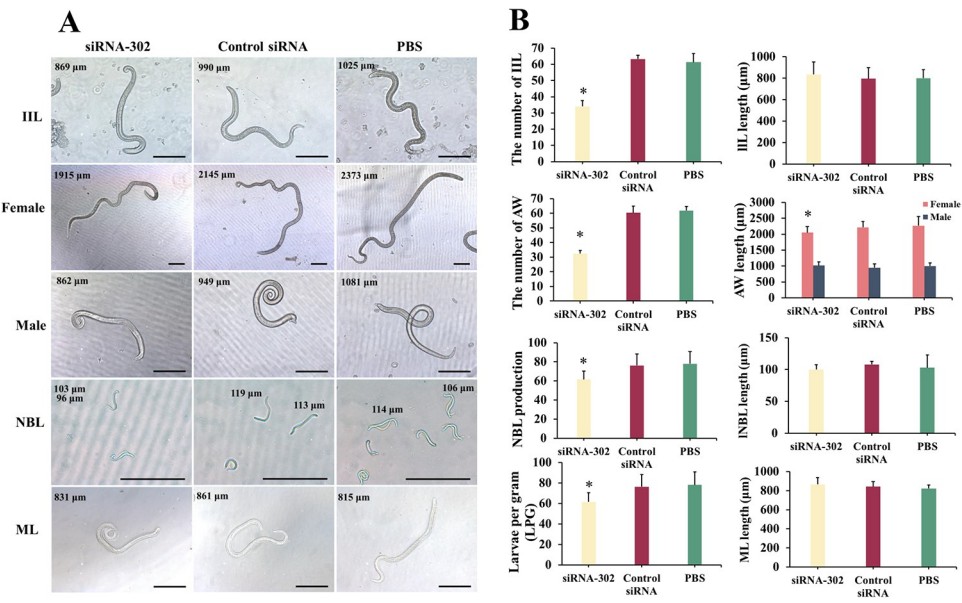

**Fig 5. Effects of RNA interference on worm burden and development in mice.** The siRNA-treated *T. spiralis* ML were used to infect mice, and worms at different developmental stages were collected for counting, observation and length measurement. Compared with the control siRNA and PBS groups, the recoveries of IIL and AW were reduced in the siRNA-302-treated group (*$P < 0.05$), the ML worm load was reduced (*$P < 0.05$), and the worm reduction rates were 44.37%, 47.57% and 57.06% for IIL, AW and ML, respectively. The female worm length and fecundity were reduced in the siRNA-302-treated group (*$P < 0.05$). Scale bar: 200 μm.

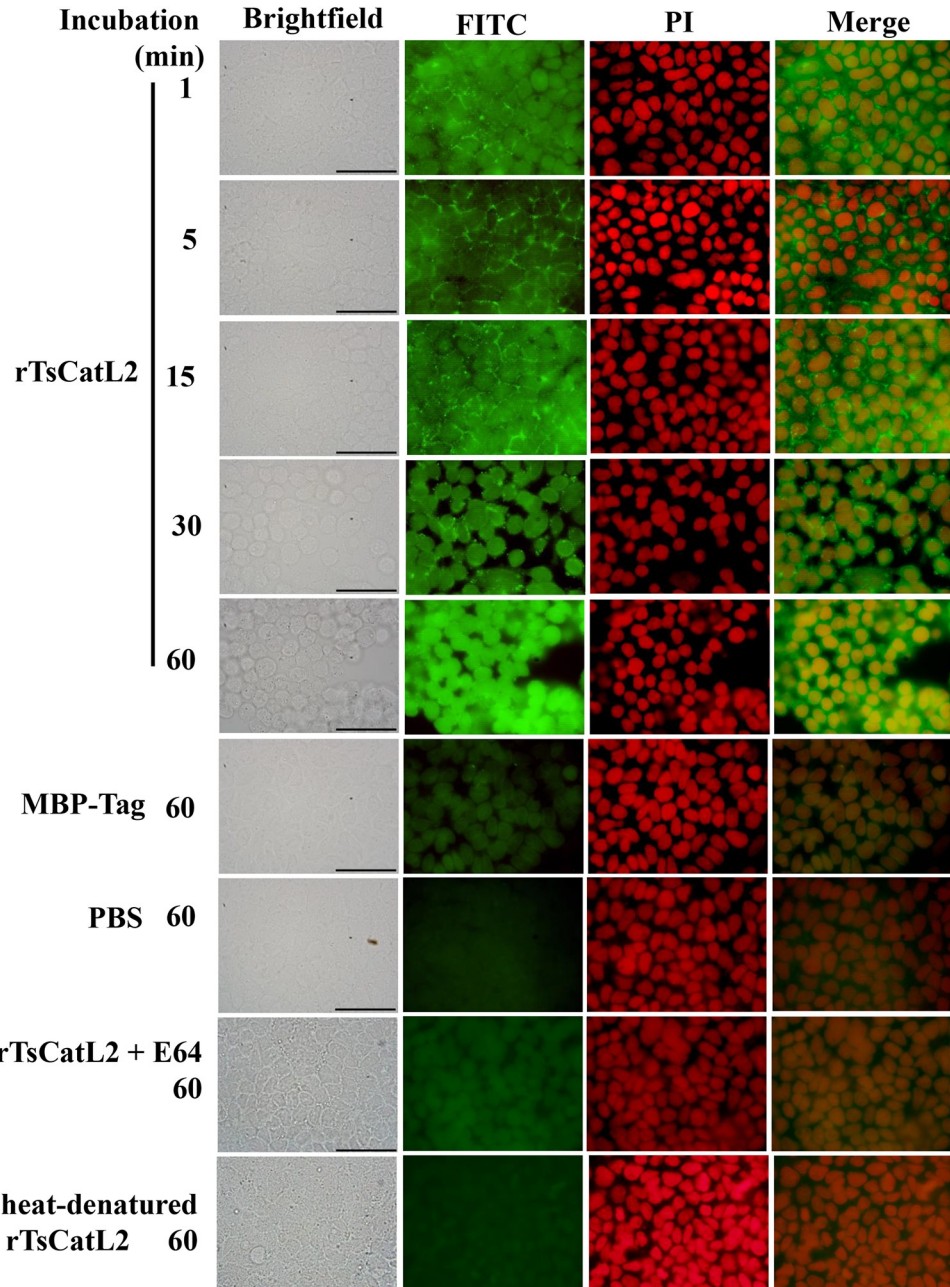

**Fig 6. rTsCatL2 binding to Caco-2 cells analyzed by fluorescence.** After co-culture of rTsCatL2 with Caco-2 cells, immunofluorescence showed that rTsCatL2 initially bound mainly around the cell periphery and gradually entered the cytoplasm with increasing incubation time. Faint fluorescence of Caco-2 cells was detected in the MBP and E64-treated rTsCatL2 groups, while no fluorescence was detected in the heat-treated rTsCatL2 group and the PBS group. Scale bar: 50 μm.

## Western blot analysis of Caco-2 cell proteins

The rTsCatL2 was added to Caco-2 cells and the cellular proteins were collected after incubation at different times. It was revealed that the expression of claudin 1 decreased by 43.23% after 30 min incubation and 58.65% after 60 min incubation ($F = 81.66$, $P < 0.05$). There was

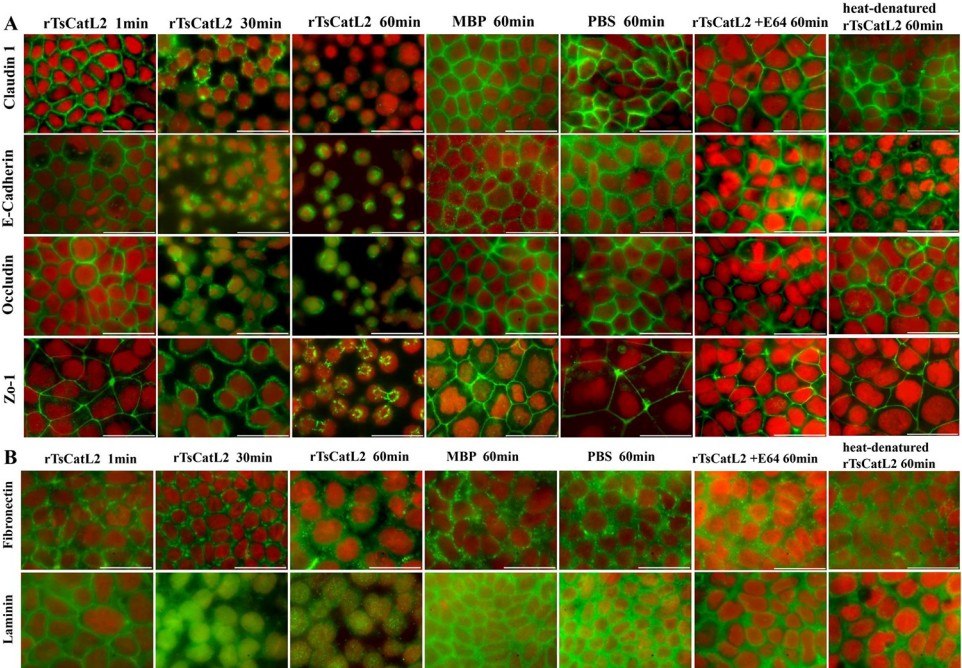

**Fig 7. Fluorescence analysis of changes in rTsCatL2 after incubation with Caco-2 cells on cell junctions (A) and extracellular matrix proteins (B).** Immunofluorescence showed that claudin 1, e-cadherin, occludin and zo-1 gradually entered the cells from the cell junctions, the fluorescence brightness of claudin 1 gradually decreased, laminin was scattered in the nucleus, while the localization of cellular proteins did not change in the MBP, PBS, E64-treated rTsCatL2, and heat-treated rTsCatL2 groups, scale bar: 50 μm.

no significant change in the expression of e-cadherin, occludin, laminin and collagen-1 protein in Caco-2 cells ($P > 0.05$) (Fig 8).

## Co-IP assay of rTsCatL2 binding to Caco-2 cellular proteins

To investigate the interaction rTsCatL2 with cellular proteins, rTsCatL2 was incubated with Caco-2 for 30 min and immunoprecipitated. rTsCatL2 was shown not to interact with claudin 1, e-cadherin, occludin and fibronectin but with laminin and collagen I (Fig 9).

## Role of rTsCatL2 on autophagy

After rTsCatL2 treatment, Caco-2 cells were stained for MDC fluorescence, and bright green fluorescence was observed upon excitation with UV light (Fig 10A). Transmission electron microscopy revealed the presence of autophagic vesicles and autophagic lysosomes in Caco-2 cells after incubation with rTsCatL2 (Fig 10B). The results indicated that rTsCatL2 induced autophagy in Caco-2 cells.

## Discussion

The cathepsin L enzyme produced by the parasite facilitates the invasion, migration, and breakdown of host proteins into absorbable nutrients [24,25]. Our previous research demonstrated that rTsCatL2 degrades immunoglobulin, hemoglobin, fibronectin, laminin and collagen I in vitro. It also exhibits the highest transcript levels in the larval stage of intestinal infection [14]. However, the role of TsCatL in *T. spiralis* invasion of host epithelial cells is unclear.

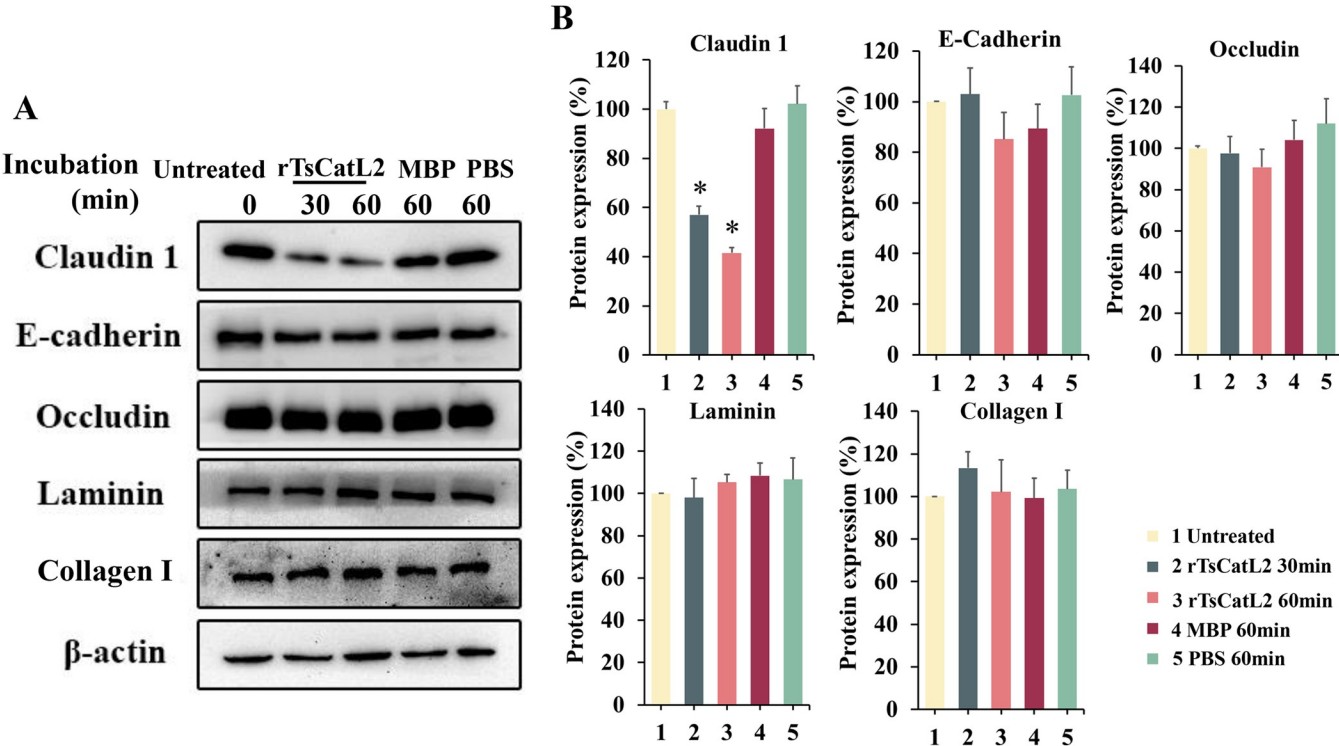

**Fig 8. Western blot analysis of cellular protein changes after incubation of rTsCatL2 with Caco-2 cells.** When rTsCatL2 was co-cultured with Caco-2 cells, the expression of claudin 1 decreased by 43.23% after 30 min incubation and by 58.65% after 60 min incubation (*$P < 0.05$), while the protein expression levels of e-cadherin, occludin, laminin and collagen-I did not change significantly ($P > 0.05$); A: ECL chemiluminescence; B: greyscale analysis.

RNAi technology has been successfully applied in various parasitic helminths, including *Brugia malayi*, *Ascaris suum*, *T. spiralis*, *Haemonchus contortus* and *Schistosoma japonicum* [26]. In studies on *T. spiralis*, silencing paramyosin using siRNA1743 reduced larval molting rate and caused severe epidermal damage [27]. Additionally, silencing serine protease, gluta-thione S-transferase reduced *T. spiralis* invasion of hosts, affecting worm development and fecundity [21,28,29]. Silencing glutaminase reduced larval infectivity, while silencing gluta-mine synthetase affected acid tolerance, molting and larval development [30,31]. In this study, three pairs of specific short-fragment siRNAs were designed to silence TsCatL by electropora-tion, and siRNA-302 was the most effective in silencing TsCatL. The specificity of siRNA-302 interference was then tested by selecting TsASP2, which has low homology to TsCatL. Western blotting showed that the siRNA-302 interference was specific and silenced only the TsCatL gene, which could be used for further experimental studies.

Cathepsin is a crucial digestive enzyme in the parasite. *Fasciola hepatica* breaks down colla-gen by secreting cathepsin L2 and cathepsin L3, which helps the worm to migrate through the host [32], and *Schistosoma japonicum* cathepsin B2, which helps the worm to invade the skin [33]. In this study, when *T. spiralis* larvae were added to Caco-2 cells, they invaded the cells, leaving a serpentine migration path without invading the control C2C12 cells. This selective invasion suggests that *T. spiralis* invasion is not purely mechanical. This result is consistent with previous studies in which *T. spiralis* readily invaded intestinal and renal epithelial cells but not fibroblasts and myocytes [34]. *T. spiralis* invaded Caco-2 cell monolayers and residues of TsCatL were detected on their migration pathways, suggesting that TsCatL was involved in the worms' invasion of epithelial cell monolayers. The addition of rTsCatL2 immune serum to

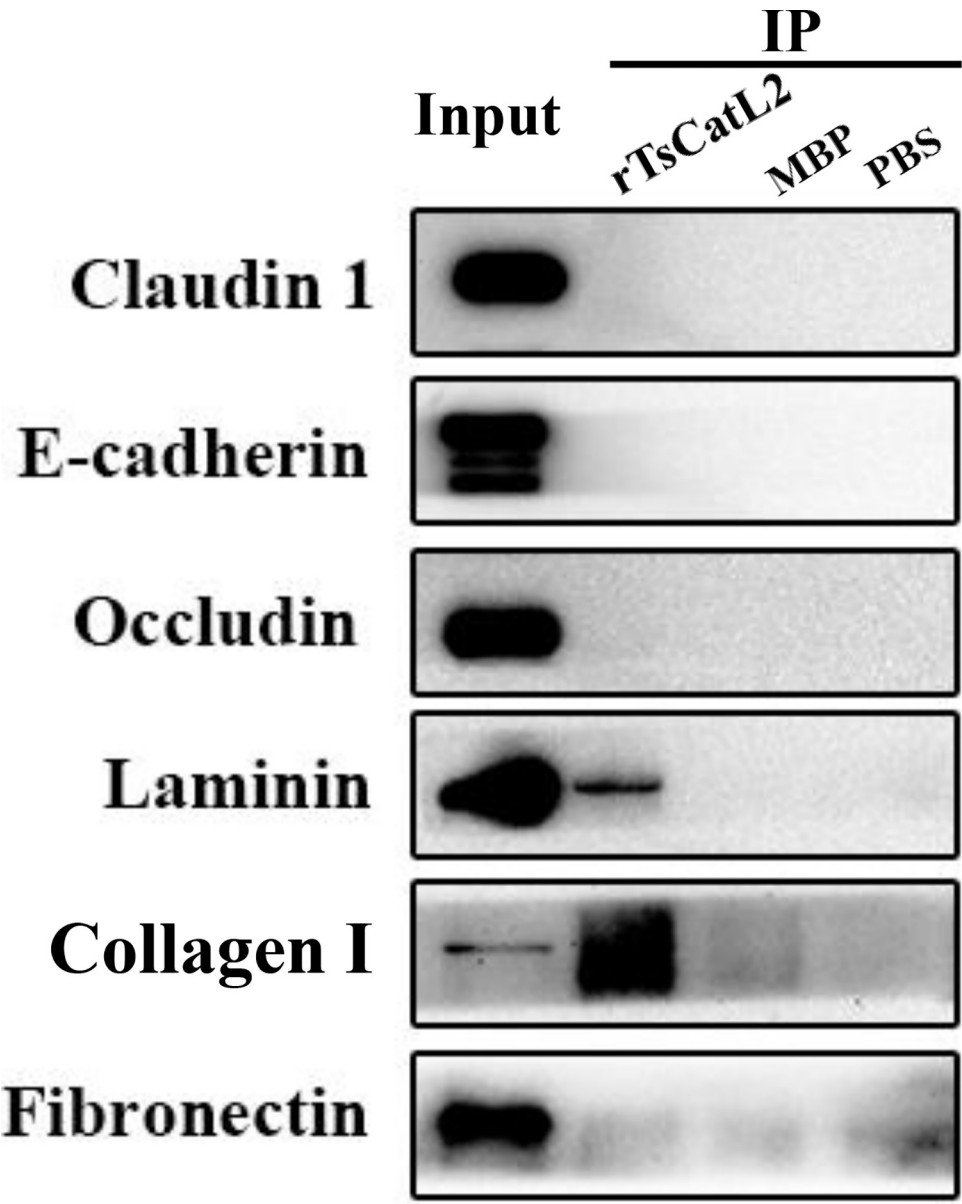

**Fig 9. Co-IP analysis of rTsCatL2 interaction with Caco-2 cell proteins.** The result showed that rTsCatL2 interacts with laminin and collagen I but not with claudin 1, e-cadherin, occludin or fibronectin.

the semi-solid medium reduced the invasion, probably because the antiserum blocked the active site of TsCatL, which in turn blocked the invasion. The reduction in larval invasion by RNAi silencing of TsCatL further suggested that TsCatL is essential for larval invasion of Caco-2 cells. Mice gavaged orally with *T. spiralis* after RNAi silencing of TsCatL showed reduced numbers of IIL and AW recovered in the intestine and reduced ML worm loads in the muscle, indicating that fewer *T. spiralis* larvae invaded the intestinal mucosa of the mice and that TsCatL was involved in the *T. spiralis* invasion of the host intestinal mucosa. Silencing of the TsCatL gene in *T. spiralis* resulted in shorter lengths of 6 d females and reduced numbers of NBL, and in conjunction with previous studies that found TsCatL localized to the embryo of the worm, it is hypothesized that TsCatL is involved in embryonic development [14].

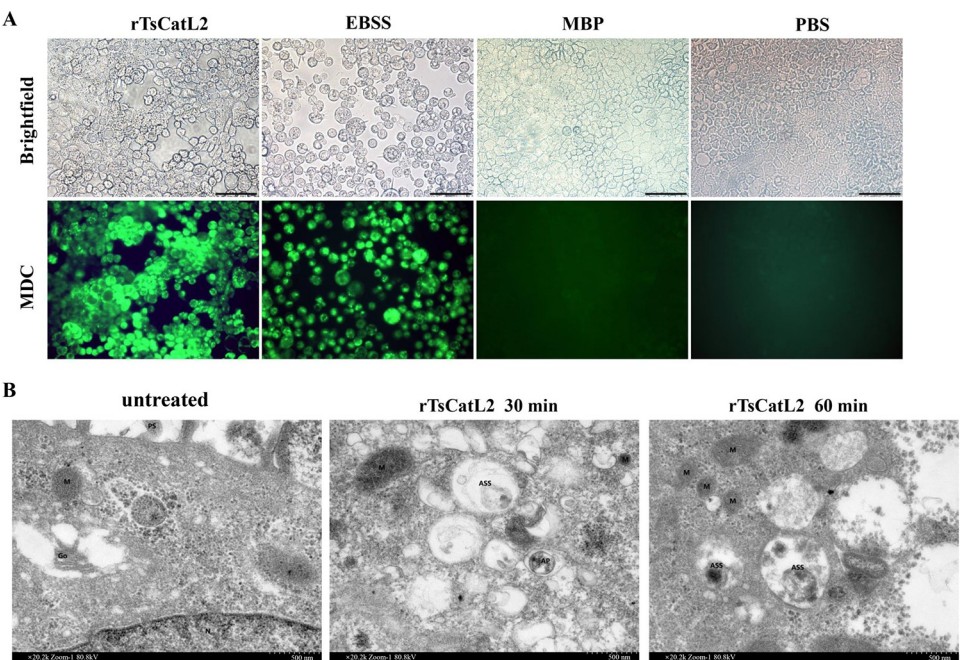

**Fig 10. Assessment of the effect of rTsCatL2 on Caco-2 autophagy.** A: MDC fluorescence staining. The rTsCatL2-treated group showed bright green fluorescence and cells underwent autophagy; the EBSS-treated positive control group underwent autophagy, while no fluorescence was observed in MBP and PBS groups, scale bar: 100 μm. B. Transmission electron microscopy. M: mitochondrion; Go: Golgi apparatus; PS: foot; ASS: autolysosome; AP: autophagosome.

Cathepsin L of *Caenorhabditis elegans* is important for embryogenesis and development [35]. Silencing the Cathepsin L of *Caenorhabditis elegans* by RNAi or mutation would result in an almost 100% embryonic lethality rate [36]. The findings showed that silencing of the TsCatL gene reduced larval invasion of host intestinal mucosa and reduced the infectivity and fecundity in the host.

The intestinal epithelium is tightly linked to form a physical barrier in the intestine, preventing pathogens from leaving intestinal lumen and invading the host. After *T. spiralis* invades the intestinal mucosa, pathological sections show edema of the intestinal mucosa, an increase in the size of the small intestinal villi, an increase in cupped cells and the size of the intestinal epithelium with a hyaline, slightly stained cytoplasm [20]. Protozoa can enter the host by breaching the intestinal barrier through virulence factors. Occludin and zo-1 levels are decreased and some microvilli on the cell surface are lost when *Toxoplasma gondii* infects Caco-2 cells [37]. Caco-2 cells infected with *Cryptosporidium parvum* increased paracellular permeability and reduced protein levels of claudin 4, occludin and e-cadherin, and C57BL/6 mice infected with *Cryptosporidium parvum* also showed significantly reduced protein expression of occludin, claudin 4 and e-cadherin in the mucosa of the ileum and jejunum [38]. The addition of *Giardia intestinalis* to Caco-2 cell monolayers resulted in changes in the integrity of cellular tight junctions, microvillar structure and extracellular mucin layer [39]. The nematodes rely on a combination of mechanical forces and virulence factors to mediate worm entry into the host intestinal mucosa. *Haemonchus contortus* make Caco-2 cell monolayers more permeable [40]. Serine and cysteine proteases in *T. spiralis* IIL down-regulate the expression of occludin, claudin 1 and e-cadherin in Caco-2 cells and up-regulate the expression of claudin 2; the specific proteins that perform this function have not been identified [10]. In vitro invasion

assays and RNAi results in *T. spiralis* demonstrated that TsCatL plays a crucial role in worm invasion of intestinal mucosa. To further analyze the reasons, rTsCatL2 was added to Caco-2 cells to analyze the effect of rTsCatL2 on cell junctions.

In this study, rTsCatL2 altered the localization of cellular tight junction proteins (claudin 1, occludin and zo-1) and adhesion-linked proteins (e-cadherin), downregulated claudin 1 expression and disrupted intestinal barrier integrity. The transmembrane protein claudin 1 is the major TJ protein in the paracellular barrier [41]. The transmembrane protein occludin is a crucial molecule regulating TJ permeability [42]. The cytoplasmic protein zo-1 is a key molecule involved in TJ assembly and signal transduction [43]. Changes in the number and localization of TJ proteins result in reduced intestinal epithelial barrier function [44].

Previous research has demonstrated that rTsCatL2 hydrolyzes host collagen I, fibronectin, and laminin in vitro, all of which are important components of the extracellular matrix (ECM), which is required for cell adhesion, signaling, migration, and metabolism [14,45]. In this study, rTsCatL2 altered the localization of laminin. However, Western blot showed no change in laminin and collagen I expression, presumably because rTsCatL2 interacts with cells for only one hour and rTsCatL2 degrades host proteins in vitro by overnight incubation [14]. Further research into rTsCatL2's interactions with Caco-2 proteins showed that rTsCatL2 interacted with collagen I and laminin but not with claudin 1, e-cadherin, occludin, or fibronectin. The results showed that the cell barrier disruption by rTsCatL2 does not rely solely on its hydrolysis but that there are other regulatory mechanisms. Studies showed that *T. spiralis* excretion/secretion proteins inhibited the expression of tight junction proteins via the MAPK signaling pathway [9]. Piezo1, a mechanosensitive and nonselective cation channel, regulates intestinal epithelial function by affecting the tight junction protein claudin-1 via the ROCK pathway [46]. L-glutamine led to an increased distribution of claudin-1 at plasma membranes in intestinal porcine epithelial cells by activating CAMKK2-AMPK signaling [47].

*Cryptosporidium parvum* was found to reduce the expression of occludin, claudin-4 and e-cadherin through autophagy, disrupting the epithelial barrier. siRNA silencing of the autophagy-associated protein ATG7 in Caco-2 cells blocked *Cryptosporidium*-induced downregulation of occludin, claudin-4 and e-cadherin [48]. Incubation of *Entamoeba histolytica* trophoblast extracts with MDCK cells in lysogenic tissue causes cell death by autophagy, leading to the destruction of the epithelial cell barrier [49]. In this study, fluorescence and projection electron microscopy results indicated that autophagy occurred in Caco-2 cells after incubation with rTsCatL2. The decrease of claudin 1 and the alteration in TJ/AJ localization may be associated with autophagy, leading to the disruption of intestinal epithelial barrier function.

In conclusion, after RNAi silenced TsCatL, the level of TsCatL transcription and expression in the worms decreased, the invasion rate of worms into Caco-2 decreased, and the development and fertility of the female worm were inhibited. Incubating rTsCatL2 with Caco-2 cells resulted in the loss of intercellular junctions, entry of cell junction proteins into the cytoplasm and cell autophagy, leading to disruption of the intestinal epithelial barrier. This study offers a novel target for the development of anti-trichinosis vaccines and drugs.

## Author Contributions

**Conceptualization:** Ruo Dan Liu, Jing Cui, Zhong Quan Wang.

**Data curation:** Ruo Dan Liu, Jing Cui, Zhong Quan Wang.

**Funding acquisition:** Ruo Dan Liu, Jing Cui.

**Investigation:** Ruo Dan Liu, Xiang Yu Meng, Chen Le Li, Xin Zhi Lin, Qiu Yi Xu, Han Xu, Shao Rong Long, Jing Cui, Zhong Quan Wang.

**Methodology:** Ruo Dan Liu, Jing Cui, Zhong Quan Wang.

**Project administration:** Ruo Dan Liu, Jing Cui, Zhong Quan Wang.

**Resources:** Ruo Dan Liu, Jing Cui, Zhong Quan Wang.

**Software:** Ruo Dan Liu, Jing Cui, Zhong Quan Wang.

**Supervision:** Ruo Dan Liu, Jing Cui, Zhong Quan Wang.

**Writing – original draft:** Ruo Dan Liu, Jing Cui, Zhong Quan Wang.

**Writing – review & editing:** Ruo Dan Liu, Jing Cui, Zhong Quan Wang.

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
