## [Decision Letter · Decision Letter 0]

1 Aug 2023

Dear Professor Cui,

Thank you very much for submitting your manuscript "Trichinella spiralis cathepsin L damages the tight junctions of intestinal epithelial cells and mediates larval invasion" for consideration at PLOS Neglected Tropical Diseases. As with all papers reviewed by the journal, your manuscript was reviewed by members of the editorial board and by several independent reviewers. In light of the reviews (below this email), we would like to invite the resubmission of a significantly-revised version that takes into account the reviewers' comments. 

The reviewers commented that this is an interesting study, however, editing and potentially other experiments are required before the manuscript is suitable for publication. Please ensure that sufficient information is provided for the methods and address all the comments raised by the reviewers.

We cannot make any decision about publication until we have seen the revised manuscript and your response to the reviewers' comments. Your revised manuscript is also likely to be sent to reviewers for further evaluation.

Sincerely,

Krystyna Cwiklinski, PhD

Academic Editor

Francesca Tamarozzi

Section Editor

The reviewers commented that this is an interesting study, however, editing and potentially other experiments are required before the manuscript is suitable for publication. Please ensure that sufficient information is provided for the methods and address all the comments raised by the reviewers.

Reviewer's Responses to Questions

**Key Review Criteria Required for Acceptance?**

**Methods**

-Are the objectives of the study clearly articulated with a clear testable hypothesis stated?

-Is the study design appropriate to address the stated objectives?

-Is the population clearly described and appropriate for the hypothesis being tested?

-Is the sample size sufficient to ensure adequate power to address the hypothesis being tested?

-Were correct statistical analysis used to support conclusions?

-Are there concerns about ethical or regulatory requirements being met?

Reviewer #1: Methods, line 136. What is the anti-rTsASP2 serum mentioned? Define this and how it was produced.

For Western blotting, how much soluble lysate was added per well? How was this quantified?

Figure 1 – how many biological replicates of the RNAi experiment have been performed? This should be clearly stated in the text/legend.

How was the % protein expression (Figure 1) quantified? More detail should be added to line 175 in the methods section.

Reviewer #2: The methods used are appropriate and the results support the conclusions.

1. Line 191,how to obtain "anti-MBP serum" should be describe clearly ;

2. Line 209~211, “rTsCatL2, MBP or PBS were ...... added to the surface of Caco-2 cells”，MBP here is protein-tag, how to obtain MBP? also expressed in E.coli and purified？ or Purchased from the company?

3. The abbreviation that first appears should indicated，such as NBL, EBSS.

Reviewer #3: The methods are well described. 

Study design could include addition of specific inhibitor such as E64 to rTsCL2 to show that inhibitor-bound cysteine protease does not affect Caco2 cells. Also, why did authors only test one concentration of enzyme - was the effect of the protease titratable?

In experiments where anti-TsCL2 was used to block parasite entry into cells - reduced at low significance - specificity could have been shown by adding rTsCL2 to the medium and demonstrating that this reversed the effect of the enzyme

**Results**

-Does the analysis presented match the analysis plan?

-Are the results clearly and completely presented?

-Are the figures (Tables, Images) of sufficient quality for clarity?

Reviewer #1: What time point post-RNAi is shown in Figure 2? Please add.

It would be of interest to see if the cellular damage shown in Figure 3 is due only to the mechanical action of worms migrating through the monolayer as whether secreted products contribute as well. Could this assay be performed with the addition of dead worms as well as T. spiralis secreted products and recombinant TsCatL2? 

Additionally, experiments with heat-denatured enzyme (or enzyme inhibited with E-64) would show whether these effects are due to active enzyme (same for later experiments; e.g. Fig 7 and the pulldown in Fig. 9).

Figure 4 – I don’t think it is necessary to plot the data as both & % invasion and % inhibition. Pick one readout for this figure.

How many individual works were measured in each group (Figure 5)?

Results describing Figure 6. It is stated in text that “Immunofluorescence with anti-rTsCatL2 antibody showed that rTsCatL2 bound mainly to the tight junctions of Caco-2 cells…”. It is not possible to say for certain what the TsCatL2 is binding to at this resolution. At best you can say green fluorescence was seen around the cell periphery. This section should be tempered to reflect this. 

Collagen is misspelt as “collage” in several places throughout the manuscript.

The labelling of the micrographs shown in Figure 10 are too small/unclear. How many cells/field of view were observed/counted to quantify autophagy at the ultrastructural level?

Reviewer #2: The results are clearly and believable，and several minor problems should be noted.

1. Line 321~324，“rTsCatL2 bound mainly to the tight junctions of Caco-2 cells..., no binding of MBP to Caco-2 cells was detected.” actually, Fig 6, MBP-tag showed faint fluorescenec，you should describe more accurately.

2. Fig 8 and Fig 9 seem to be contradictory. Fig 8, the rTsCatL2 was added to Caco-2 cells, western blot showed no change in laminin and collagen I expression; and Fig 9, showed rTsCatL2 did not interact with claudin 1, e-cadherin, occludin and fibronectin but with collagen I and laminin. How to explain?

Reviewer #3: The amount of knockdown by iRNA is shown to be specific and significant but is nevertheless low - less than 50% with all iRNAs and for transcripts and protein -and it does not add confidence that parasites with only this reduction would be necessarily damaged and lack viability.

**Conclusions**

-Are the conclusions supported by the data presented?

-Are the limitations of analysis clearly described?

-Do the authors discuss how these data can be helpful to advance our understanding of the topic under study?

-Is public health relevance addressed?

Reviewer #1: Since the silencing/antibody blocking of TsCatL2 was only partially successful at blocking entry of the worms into the epithelial cells, could the authors please speculate on other enzymes that may be compensating and the implications of this for parasite control.

Reviewer #2: (No Response)

Reviewer #3: I feel that most experiments do not achieve strong significance in parasite invasion or infectivity and additional experiments as suggested above would strengthen the data

**Editorial and Data Presentation Modifications?**

Reviewer #1: Line 358 – previous research?

Reviewer #2: Minor revision. The quality of English needs improving, I suggest that you obtain assistance from a colleague whose native language is English.

Reviewer #3: (No Response)

**Summary and General Comments**

Reviewer #1: In this study the authors aimed to investigate the role of Trichinella spiralis cathepsin L (TsCatL2) during penetration of intestinal epithelial cells. Invasion of cells in vitro, as well as establishment of infection of mice, were both reduced when TsCatL2 was silenced using RNAi. They propose a mechanism for this where TsCatL2 disrupts the barrier function of the intestine by degradation of extracellular matrix/cell junction proteins.

Specific comments:

Introduction, line 106. Replace “substance” with “cysteine peptidase” to describe Giardipain-1 here.

Introduction, line 109. Explain what domains TsCatL2 refers to – was only the mature enzyme domain cloned and expressed as a recombinant? 

Do the authors have any information on the transcriptional profile of TsCatL? Is it specifically upregulated to coincide with arrival of the worms in the intestine? This should be included if known.

Line 376 – it should be made clear that Fasciola hepatica secretes a large family of cathepsins L, only some of which (primarily FhCL2 and FhCL3) have collagenolytic activity.

The section of ECM/cell junctions in the discussion was overlong and could be made more concise.

Suggested further experiments are suggested in the results comments.

Reviewer #2: The topic is novel and interesting, and the methods used are appropriate. "MBP" I mentioned above is important, for rTsCatL2 were expressed in Escherichia coli, recombinant protein of Prokaryotic expression may contain endotoxins，it should avoid the effect of endotoxins on Caco-2 cell junctions, when you observe the effect of rTsCatL2 on Caco-2 cell junctions.

Reviewer #3: The data could be improved by better experimental design and control experiments as suggested above e.g. titrating protease, using specific inhibitors, uisng reverse inhibtion of antibodies.

PLOS authors have the option to publish the peer review history of their article (what does this mean?). If published, this will include your full peer review and any attached files.

Reviewer #1: No

Reviewer #2: No

Reviewer #3: No
---

## [Decision Letter · Decision Letter 1]

1 Nov 2023

Dear Professor Cui,

Thank you very much for submitting your manuscript "Trichinella spiralis cathepsin L damages the tight junctions of intestinal epithelial cells and mediates larval invasion" for consideration at PLOS Neglected Tropical Diseases. As with all papers reviewed by the journal, your manuscript was reviewed by members of the editorial board and by several independent reviewers. The reviewers appreciated the attention to an important topic. Based on the reviews, we are likely to accept this manuscript for publication, providing that you modify the manuscript according to the review recommendations. 

The authors have addressed the reviewers comments, however some minor points still need addressing before the manuscript is suitable for publication.

Sincerely,

Krystyna Cwiklinski, PhD

Academic Editor

Francesca Tamarozzi

Section Editor

The authors have addressed the reviewers comments, however some minor points still need addressing before the manuscript is suitable for publication.

Reviewer's Responses to Questions

**Key Review Criteria Required for Acceptance?**

**Methods**

-Are the objectives of the study clearly articulated with a clear testable hypothesis stated?

-Is the study design appropriate to address the stated objectives?

-Is the population clearly described and appropriate for the hypothesis being tested?

-Is the sample size sufficient to ensure adequate power to address the hypothesis being tested?

-Were correct statistical analysis used to support conclusions?

-Are there concerns about ethical or regulatory requirements being met?

Reviewer #1: (No Response)

Reviewer #2: 1. the sequence of TsCatL2 should have signal peptide, when construct pMAL-c2X/TsCatL2 recombinant plasmid, contained the signal peptide ?

Reviewer #3: Authors have addressed my concerns and improved figures as suggested.

**Results**

-Does the analysis presented match the analysis plan?

-Are the results clearly and completely presented?

-Are the figures (Tables, Images) of sufficient quality for clarity?

Reviewer #1: (No Response)

Reviewer #2: 1. Line 308，“in the TsCatL siRNA-487”， and line 310 "lower in the siRNA-487 group than in the control siRNA...Fig 4B", I'm not find siRNA-487 in the methods, and Fig 4B showed siRNA-302, but the legend is siRNA-487 group, maybe an error?

2. legend of Fig4, "the rate of worm invasion was lower in the siRNA-487 group", the picture showed inhibition%, be consistent.

3. Fig 6 and Fig 7, rTsCatL2,MBP, rTsCatL2+E64, incubation with Caco-2 cells within 60min, your response mentioned "because adding rTsCatL2 to the liquid medium severely damaged the Caco-2 cell monolayer within 1 hour", if co-culture of MBP and rTsCatL2+E64 with Caco-2 cells more than 1h, whether the similar appearance ?

Reviewer #3: Yes

**Conclusions**

-Are the conclusions supported by the data presented?

-Are the limitations of analysis clearly described?

-Do the authors discuss how these data can be helpful to advance our understanding of the topic under study?

-Is public health relevance addressed?

Reviewer #1: (No Response)

Reviewer #2: (No Response)

Reviewer #3: Yes

**Editorial and Data Presentation Modifications?**

Reviewer #1: (No Response)

Reviewer #2: (No Response)

Reviewer #3: None

**Summary and General Comments**

Reviewer #1: (No Response)

Reviewer #2: rTsCatL2 downregulated claudin 1 expression, but not interacted with claudin 1, what's the possible mechanisms, should further discussed.

Reviewer #3: Paper is much improved

PLOS authors have the option to publish the peer review history of their article (what does this mean?). If published, this will include your full peer review and any attached files.

Reviewer #1: No

Reviewer #2: No

Reviewer #3: No

Figure Files:

Data Requirements:

Reproducibility:

References

---

## [Editor Report · Decision Letter 2]

22 Nov 2023

Dear Professor Cui,

We are pleased to inform you that your manuscript 'Trichinella spiralis cathepsin L damages the tight junctions of intestinal epithelial cells and mediates larval invasion' has been provisionally accepted for publication in PLOS Neglected Tropical Diseases.

Best regards,

Krystyna Cwiklinski, PhD

Academic Editor

Francesca Tamarozzi

Section Editor

The authors have addressed the comments raised by the reviewers. The manuscript is now suitable for publication in PLoS NTD.

---

## [Editor Report · Acceptance letter]

25 Nov 2023

Dear Professor Cui,

We are delighted to inform you that your manuscript, "*Trichinella spiralis* cathepsin L damages the tight junctions of intestinal epithelial cells and mediates larval invasion," has been formally accepted for publication in PLOS Neglected Tropical Diseases.

Best regards,

Shaden Kamhawi

co-Editor-in-Chief

Paul Brindley

co-Editor-in-Chief
